# Learning Expressive Representations with Kolmogorov-Arnold Network Autoencoders

## Abstract

Processing high-dimensional data presents significant challenges, including the curse of dimensionality and the limitations of models with static, non-adaptive components. While autoencoders are a cornerstone of unsupervised representation learning, their performance is often constrained by traditional activation functions. Inspired by the Kolmogorov-Arnold representation theorem, this paper introduces Kolmogorov-Arnold Network Autoencoders (KAN-AEs), a novel framework that replaces static activations with learnable, spline-based functions. We further propose a Convolutional KAN-AE (CKAN-AE) variant that incorporates spatial inductive biases for image data. Through comprehensive experiments on benchmark datasets, we demonstrate that KAN-based autoencoders consistently achieve superior reconstruction fidelity and learn more discriminative latent representations, as evidenced by improved linear probing accuracy. Notably, CKAN-AE excels on complex natural images. While the enhanced expressiveness introduces a computational trade-off, our work establishes KAN-AEs as a powerful tool for scenarios where representation quality is paramount, paving the way for more adaptive and efficient deep learning models.

## 1 Introduction

In the age of big data, machine learning algorithms are increasingly tasked with processing and interpreting high-dimensional datasets across a variety of domains (LeCun et al., 2015; He et al., 2016; Schmidhuber, 2015). The ability to extract meaningful insights from such data has fueled advancements in fields as diverse as genomics, climate science, and financial analytics (Hassan et al., 2022; Fathi et al., 2022; Sohangir et al., 2018). While high-dimensional data offers rich informational potential, it also introduces significant challenges, including computational complexity, large storage requirements, and the heightened risk of overfitting (Bolón-Canedo et al., 2015; Tufail et al., 2023). These challenges are often exacerbated by the "curse of dimensionality," a phenomenon wherein the effective volume of the feature space grows exponentially with the dimensionality, leading to sparsity and diminishing model performance (Verleysen & François, 2005).

To mitigate these issues, the machine learning field has seen the development of advanced techniques aimed at reducing dimensionality and learning efficient representations. Among these, autoencoders have emerged as a fundamental tool in unsupervised learning, capable of capturing the underlying structure of data in a compressed form (Hinton & Salakhutdinov, 2006; Bengio et al., 2013). However, traditional autoencoder architectures often rely on static activation functions such as ReLU or sigmoid, which may limit their adaptability to the complex geometries and dynamics inherent in high-dimensional data.

Recent advancements have sought to address these limitations by innovating network architectures and activation functions. Notably, the success of Kolmogorov-Arnold Networks (KANs) (Liu et al., 2025), inspired by the Kolmogorov-Arnold representation theorem, has highlighted the potential of neural networks to approximate high-dimensional mappings with enhanced expressiveness. Building on these innovations, learnable activation functions such as Swish (Ramachandran et al., 2017) and Mish (Misra, 2019) have further expanded the capacity of neural networks to adapt to varying data distributions.

Building on these advances, we propose the Kolmogorov-Arnold Network Autoencoder (KAN-AE) and its Convolutional KAN-AE (CKAN-AE) variant. We investigate their potential for efficient and expressive

representation learning, hypothesizing that their inherently adaptive, function-learning layers are particularly well-suited for capturing the intricate structures in high-dimensional domains such as image data.

## 2 Related Work

The study of KANs has gained significant attention in recent years as a promising alternative to traditional multilayer perceptrons (MLPs) (Samadi et al., 2024). Unlike conventional MLPs, which employ static linear weights followed by fixed activation functions, KANs replace linear weights with learnable activation functions, allowing for dynamic pattern learning and enhanced adaptability. This architectural innovation has enabled KANs to achieve superior performance with fewer parameters, outperforming larger MLPs in terms of accuracy, scaling efficiency, and interpretability (Vaca-Rubio et al., 2024). These characteristics make KANs particularly attractive for applications requiring compact yet expressive models.

KANs have demonstrated significant success across various domains. In graph learning, for example, specialized KAN-based architectures such as the Kolmogorov-Arnold Graph Isomorphism Network (KAGIN) and the Kolmogorov-Arnold Graph Convolution Network (KAGCN) have achieved state-of-the-art results in graph regression tasks by providing more effective node feature updates (Bresson et al., 2024). These models leverage the flexibility of learnable activation functions to capture intricate relationships within graph structures, surpassing the capabilities of traditional MLPs. Similarly, KANs have been shown to improve transfer learning frameworks by replacing conventional linear probing layers in ResNet-50 architectures with KAN layers (Shen & Younes, 2024), significantly enhancing adaptability to complex and diverse data patterns while improving generalization performance.

KAN principles have also been successfully integrated into Convolutional Neural Networks (CNNs). For instance, the Residual KAN (RKAN) incorporates KAN modules into established architectures like ResNet and DenseNet. By using Chebyshev polynomial-based convolutions, RKAN achieves improved feature extraction capabilities while maintaining computational efficiency (Yu et al., 2024).

More broadly, the success of KANs is part of a larger research theme focused on learnable activation functions to enhance network adaptability. This line of inquiry includes methods such as the Parametric Rectified Linear Unit (PReLU) (He et al., 2015) and various spline-based activation functions (Bohra et al., 2020), all of which allow activation characteristics to be optimized during training.

Our work directly builds upon these advancements. While prior research has successfully applied KANs to discriminative tasks and integrated them into convolutional blocks, their potential for unsupervised representation learning via autoencoders remains largely unexplored. This paper bridges that gap by proposing KAN-AEs. By integrating the adaptive, learnable function paradigm of KANs into the autoencoder framework, KAN-AEs aim to provide a more powerful and flexible solution for core unsupervised tasks such as dimensionality reduction and data reconstruction.

## 3 Method

### 3.1 Architecture

KAN-AEs employ a symmetric encoder-decoder structure to learn efficient data representations. As illustrated in Figure 1, the encoder compresses high-dimensional input data into a lower-dimensional latent space, while the decoder reconstructs the original input from this compressed representation.

The encoder maps high-dimensional input data $\mathbf{x} \in \mathbb{R}^D$ into a compressed, lower-dimensional latent space $\mathbf{z} \in \mathbb{R}^d$, where $d \ll D$. This is achieved through a composition of functions, each corresponding to a layer in the encoder:

$$\mathbf{z} = (\Phi_{L-1} \circ \Phi_{L-2} \circ \cdots \circ \Phi_0)(\mathbf{x}), \tag{1}$$

where each $\Phi_i$ represents a transformation applied at layer $i$, typically combining a linear transformation, a learnable activation function, and optional normalization techniques. This layered approach progressively encodes the input $\mathbf{x}$ into increasingly abstract representations, culminating in the latent variable $\mathbf{z}$, which captures the most salient features necessary for reconstruction.

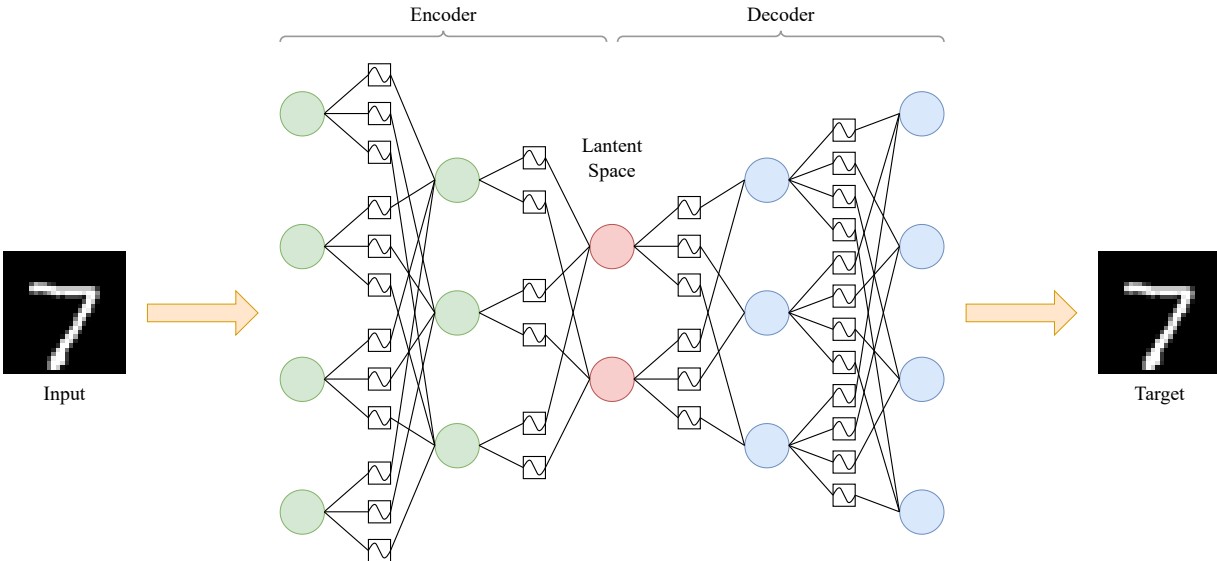

Figure 1: Overview of the KAN-AE architecture.

The decoder, conversely, reconstructs the original high-dimensional input $\mathbf{x}$ from its latent representation $\mathbf{z}$. This is achieved through another sequence of function compositions, reversing the encoding process:

$$\hat{\mathbf{x}} = (\Psi_{L-1} \circ \Psi_{L-2} \circ \cdots \circ \Psi_0)(\mathbf{z}), \tag{2}$$

where each $\Psi_i$ represents a transformation applied at layer $i$ of the decoder. These transformations progressively map the latent representation $\mathbf{z}$ back into the original data space. The output $\hat{\mathbf{x}}$ is the reconstruction of the input data $\mathbf{x}$, with the aim of minimizing the discrepancy between $\hat{\mathbf{x}}$ and $\mathbf{x}$.

### 3.2 KAN Layer

The Kolmogorov-Arnold representation theorem establishes that any continuous function mapping from $[0, 1]^n$ to $\mathbb{R}$ can be decomposed into a sum of univariate continuous functions combined through addition operations (Schmidt-Hieber, 2021). This theorem serves as a cornerstone in high-dimensional function approximation and is formally expressed as:

$$f(\mathbf{x}) = f(x_1, \ldots, x_n) = \sum_{q=1}^{2n+1} \psi_q \left( \sum_{p=1}^{n} \phi_{q,p}(x_p) \right), \tag{3}$$

where $\psi_{q,p} : [0, 1] \to \mathbb{R}$ and $\Phi_q : \mathbb{R} \to \mathbb{R}$ are continuous univariate functions.

Inspired by this theorem, a KAN layer replaces the static linear weights and fixed activations of a traditional layer with a set of learnable univariate functions. For an input vector $\mathbf{x} \in \mathbb{R}^n$, the output of a KAN layer with $m$ neurons is computed as:

$$\Phi(\mathbf{x}) = \sum_{i=1}^{N} \phi_i(x_i), \tag{4}$$

where each $\phi_i$ is a distinct, parameterized univariate function. This allows the layer to learn highly adaptive, non-linear transformations tailored to the data.

Each univariate function $\phi(x)$ is implemented as a flexible, learnable module combining a spline and a base activation function:

$$\phi(x) = \sum_{i=1}^{l+p} B_{i,p}(x) \cdot w_i + b(x) \cdot w_0, \tag{5}$$

where $B_{i,p}(x)$ are B-spline basis functions of degree $p$, which provide a localized and adaptable basis for approximating functions. The grid size, $l$, determines the number of basis functions, and $w_i$ are the trainable coefficients that adjust the contribution of each basis function. The term $b(x)$ represents the base activation function, which is scaled by the trainable coefficient $w_0$.

### 3.3 Convolutional KAN Layer

To process image data efficiently, we extend the KAN principle to a convolutional operation, leveraging the concept of Convolutional KANs (CKANs) (Bodner et al., 2024) to create CKAN layers. This design introduces translation equivariance and parameter sharing, drastically improving parameter efficiency over a fully-connected KAN layer applied to image patches.

A CKAN layer are defined by a kernel of learnable univariate functions. For a kernel of size $n \times m$, we have:

$$\Phi = \begin{bmatrix} \phi_{1,1} & \phi_{1,2} & \phi_{1,3} & \cdots & \phi_{1,m} \\ \phi_{2,1} & \phi_{2,2} & \phi_{2,3} & \cdots & \phi_{2,m} \\ \phi_{3,1} & \phi_{3,2} & \phi_{3,3} & \cdots & \phi_{3,m} \\ \vdots & \vdots & \vdots & \ddots & \vdots \\ \phi_{n,1} & \phi_{n,2} & \phi_{n,3} & \cdots & \phi_{n,m} \end{bmatrix}, \tag{6}$$

where each $\phi_{i,j}$ is a parameterized univariate function. This kernel is convolved across the spatial dimensions of the input feature map. At each location, the input values covered by the kernel are passed through their corresponding functions $\phi_{i,j}$, and the results are summed to produce the output activation, mirroring the standard convolution operation but with learnable non-linearities.

For the decoder, we implement corresponding Transposed CKAN (TCKAN) layers to perform learnable upsampling and spatial expansion. By stacking CKAN layers in the encoder and TCKAN layers in the decoder, we construct a fully convolutional KAN-based autoencoder, which we term a CKAN-AE.

## 4 Experiments

### 4.1 Experimental Setup

**Self-supervised pretraining.** During the self-supervised pretraining stage, the model is trained over 50 epochs to learn meaningful and robust representations in the latent space. We used the AdamW optimizer (Loshchilov, 2017) with a learning rate of $1 \times 10^{-2}$ for MNIST (Deng, 2012) and Fashion-MNIST (Xiao et al., 2017) and $2 \times 10^{-3}$ for CIFAR-10 and STL-10. All models were trained with a weight decay of $1 \times 10^{-4}$, a cosine annealing learning rate schedule (Loshchilov & Hutter, 2016), and a batch size of 1024 to optimize reconstruction loss. Unless stated otherwise, the latent dimension was fixed at 16.

**Linear probing.** The linear probing phase evaluates the quality of the representations learned during pretraining. Here, the pretrained encoder is frozen, and a lightweight linear classifier is trained on the extracted latent features. The training is conducted over 5 epochs using the SGD optimizer (Robbins & Monro, 1951) with a learning rate of 0.1, momentum (Qian, 1999) set to 0.9 , and a batch size of 256. This stage provides a quantitative assessment of the discriminative power of the learned features.

### 4.2 Quantitative Results

We evaluated models using two key metrics: reconstruction fidelity (Mean Squared Error, MSE) and representation quality (linear probing accuracy). Our proposed KAN-AE and CKAN-AE were compared against two baseline autoencoders: a Multilayer Perceptron Autoencoder (MLP-AE) and a CNN Autoencoder (CNN-AE). All results are reported as the mean and standard deviation across three random seeds. Table 1 presents results on simpler, grayscale datasets, while Table 2 shows results on more complex, natural image datasets.

Table 1 demonstrates that both KAN-AE and CKAN-AE consistently outperform MLP-AE and CNN-AE on grayscale datasets, achieving significantly lower reconstruction error and higher linear probing accuracy.

| Model | Params | FLOPs | MNIST | | Fashion-MNIST | |
|---|---|---|---|---|---|---|
| | | | MSE | Accuracy (%) | MSE | Accuracy (%) |
| MLP-AE | 410K | 563K | $0.0331 \pm 0.0085$ | $69.47 \pm 10.72$ | $0.0214 \pm 0.0010$ | $70.20 \pm 1.50$ |
| CNN-AE | 402K | 627K | $0.0217 \pm 0.0035$ | $83.47 \pm 2.08$ | $0.0188 \pm 0.0013$ | $76.20 \pm 2.56$ |
| KAN-AE | 409K | 563K | $0.0153 \pm 0.0001$ | $89.70 \pm 0.40$ | $0.0138 \pm 0.0002$ | $78.03 \pm 0.31$ |
| CKAN-AE | 402K | 911K | $\mathbf{0.0119} \pm 0.0001$ | $\mathbf{90.37} \pm 0.23$ | $\mathbf{0.0124} \pm 0.0001$ | $\mathbf{78.80} \pm 0.35$ |

Table 1: Performance comparison on MNIST and Fashion-MNIST datasets. Results show mean $\pm$ standard deviation over three random seeds.

| Model | Params | FLOPs | CIFAR-10 | | STL-10 | |
|---|---|---|---|---|---|---|
| | | | MSE | Accuracy (%) | MSE | Accuracy (%) |
| MLP-AE | 1.58M | 1.58M | $0.0274 \pm 0.0044$ | $26.87 \pm 3.44$ | $0.0225 \pm 0.0012$ | $27.97 \pm 0.25$ |
| CNN-AE | 1.58M | 4.23M | $0.0175 \pm 0.0002$ | $30.27 \pm 1.69$ | $\mathbf{0.0163} \pm 0.0002$ | $29.97 \pm 3.65$ |
| KAN-AE | 1.58M | 2.17M | $0.0192 \pm 0.0007$ | $31.07 \pm 0.40$ | $0.0168 \pm 0.0001$ | $29.23 \pm 1.62$ |
| CKAN-AE | 1.58M | 5.12M | $\mathbf{0.0174} \pm 0.0001$ | $\mathbf{32.73} \pm 0.57$ | $\mathbf{0.0163} \pm 0.0001$ | $\mathbf{31.00} \pm 0.50$ |

Table 2: Performance comparison on CIFAR-10 and STL-10 datasets. Results show mean $\pm$ standard deviation over three random seeds.

Notably, CKAN-AE delivers the best performance overall, highlighting the advantage of the convolutional inductive bias for image data.

On the more challenging CIFAR-10 and STL-10 datasets (Table 2), CKAN-AE again achieves the best or competitive reconstruction fidelity and the highest linear probing accuracy. This result confirms that the learnable, spatially-aware transformations of CKAN layers are particularly beneficial for capturing the complex structures in natural images.

### 4.3 Qualitative Analysis

**Reconstructed images.** Figure 2 provides a visual comparison of original and reconstructed datasets. KAN-AE and CKAN-AE produce sharper, more faithful reconstructions than the MLP-AE baseline, effectively preserving fine details and complex patterns.

**Latent space interpolation.** To evaluate the quality of the learned latent space, we performed linear interpolations between selected data points in this space. Given two latent vectors, $\mathbf{z}_0 = f(\mathbf{x}_0)$ and $\mathbf{z}_1 = f(\mathbf{x}_1)$, where $f(\cdot)$ is the encoder function, we compute a linear interpolation, $\mathbf{z}_\lambda = (1 - \lambda)\mathbf{z}_0 + \lambda\mathbf{z}_1$, where $\lambda \in [0, 1]$. Decoding these interpolated latent vectors with the decoder function $g(\cdot)$ produces smooth transitions between the original data points, $\hat{\mathbf{x}}_\lambda = g(\mathbf{z}_\lambda)$. The results, with $\lambda$ chosen at equal intervals, shown in Figure 3, demonstrate the model's ability to learn continuous, meaningful, and semantically coherent representations within its latent space.

### 4.4 Ablation Studies

To understand the impact of key architectural and design choices, we performed ablation studies on the MNIST dataset.

**Encoder and decoder design.** The full CKAN-AE architecture, which uses CKAN layers in the encoder and TCKAN layers in the decoder, yields the best performance. Substituting either component with standard MLP layers leads to a measurable drop in both reconstruction fidelity and representation quality.

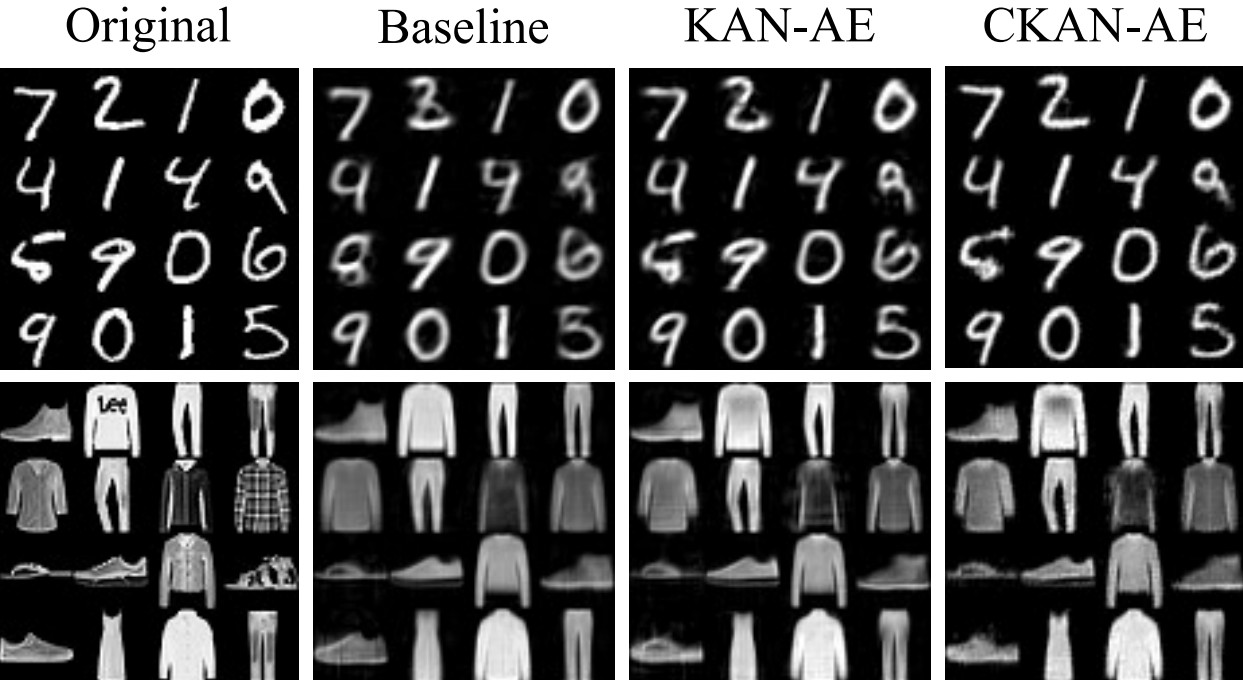

Figure 2: Visual comparison of original and reconstructed images on MNIST and Fashion-MNIST datasets.

| Encoder | Decoder | MSE | Accuracy (%) |
|---------|---------|-----|--------------|
| MLP | TCKAN | $0.0184 \pm 0.0032$ | $84.40 \pm 3.32$ |
| CKAN | MLP | $0.0134 \pm 0.0002$ | $89.80 \pm 0.46$ |
| CKAN | TCKAN | $\mathbf{0.0119} \pm 0.0001$ | $\mathbf{90.37} \pm 0.23$ |

Table 3: Ablation study on encoder and decoder design. Default settings are highlighted in gray .

**Loss functions.** We evaluated four common reconstruction losses. While performance differences are minor, MSE and Normalized Root MSE (NRMSE) offer the best balance between low reconstruction error and high linear probing accuracy.

**Grid size.** The grid size controls the number of B-spline basis functions and thus the expressivity of each learnable activation. As shown in Table 5, increasing grid size generally improves reconstruction (lower MSE) at the cost of more parameters, with diminishing returns beyond grid size of 3. A grid size of 3 provides a favorable trade-off between model capacity and parameter efficiency, and is used as our default.

## 5 Discussion

Our findings demonstrate that the KAN-AE framework, through its integration of Kolmogorov-Arnold network layers, effectively addresses key challenges in processing high-dimensional data. By employing learnable, spline-based activation functions, KAN-AE and its convolutional counterpart, CKAN-AE, enable more dynamic and adaptive feature transformations. This architectural innovation leads to superior reconstruction fidelity and, critically, to latent representations of higher quality, as evidenced by improved linear probing accuracy across diverse datasets. The comprehensive qualitative and quantitative analyses confirm the models' enhanced ability to capture intricate patterns and semantic structures.

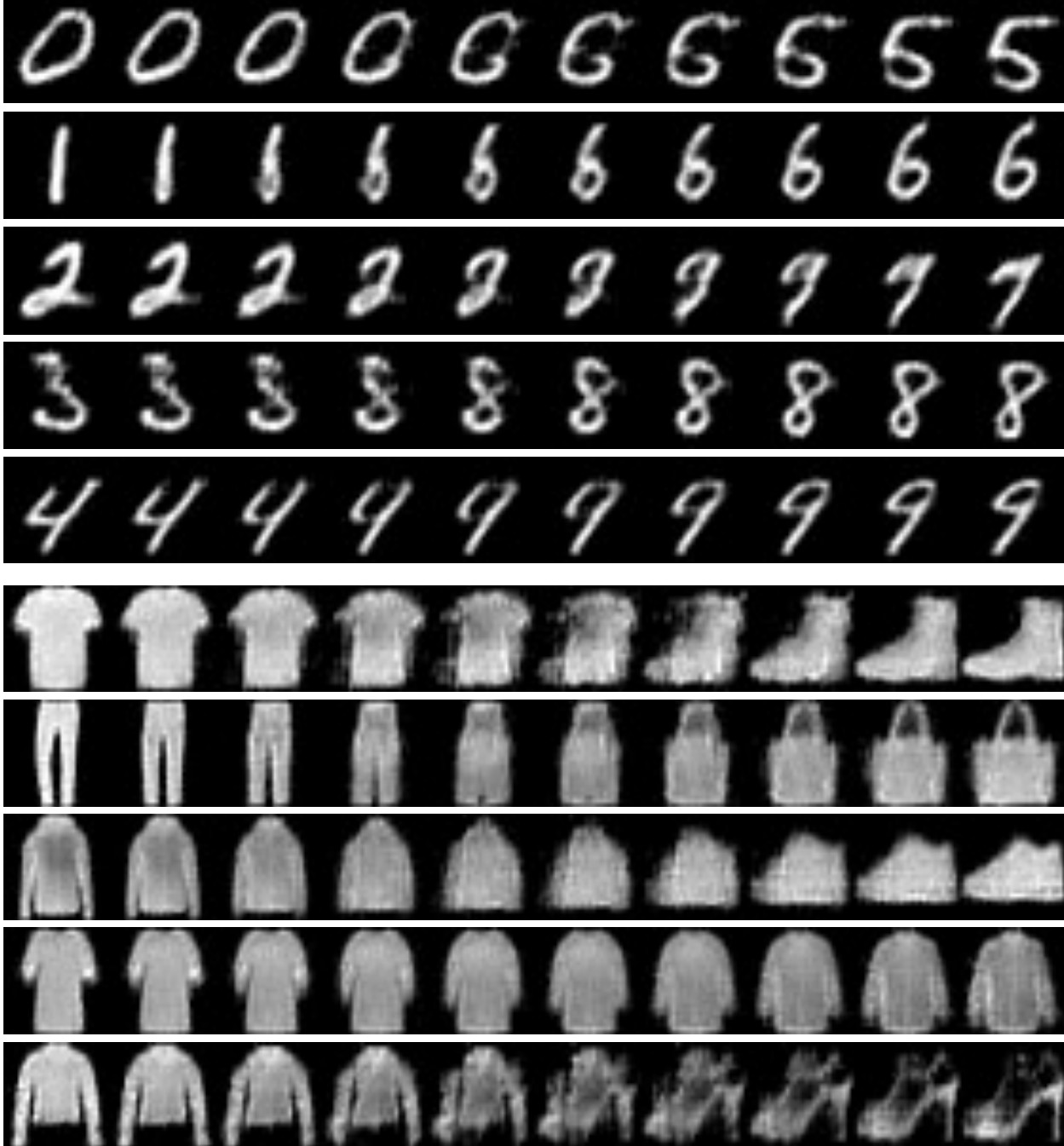

Figure 3: Visualization of linear interpolation between latent representations on MNIST and Fashion-MNIST datasets.

A primary consideration, however, is the computational overhead introduced by the spline-based activations, especially within the convolutional layers of CKAN-AE. CKAN-AE usually incurs higher FLOPs than its standard CNN-AE counterpart with similar parameter count. This increased complexity can impact training time and inference latency, potentially limiting immediate scalability for real-time applications or exceptionally high-dimensional domains. Nevertheless, for tasks where representation quality and reconstruction precision are paramount—such as in scientific data analysis or detailed generative modeling—the performance gains offered by KAN-based autoencoders present a compelling trade-off.

| Type | MSE | Accuracy (%) |
|------|-----|--------------|
| Log Cosh | $0.0121 \pm 0.0001$ | $90.40 \pm 0.40$ |
| MAE | $0.0155 \pm 0.0006$ | $89.03 \pm 0.38$ |
| MSE | $\mathbf{0.0119} \pm 0.0001$ | $90.37 \pm 0.23$ |
| NRMSE | $0.0120 \pm 0.0003$ | $\mathbf{90.43} \pm 0.25$ |

Table 4: Ablation study on loss functions. Default settings are highlighted in gray .

| Size | Params | MSE | Accuracy (%) |
|------|--------|-----|--------------|
| 1 | 301K | $0.0140 \pm 0.0003$ | $90.20 \pm 0.36$ |
| 2 | 352K | $0.0126 \pm 0.0005$ | $\mathbf{90.67} \pm 0.68$ |
| 3 | 402K | $0.0119 \pm 0.0001$ | $90.37 \pm 0.23$ |
| 4 | 452K | $\mathbf{0.0116} \pm 0.0002$ | $89.77 \pm 0.38$ |

Table 5: Ablation study on B-spline grid size. Default settings are highlighted in gray .

Future research should pursue several promising directions to advance this paradigm. First, exploring hybrid architectures that strategically place KAN/CKAN layers within deeper networks could optimize the performance-efficiency balance. Second, developing more efficient approximations of the spline computations or alternative parameterizations for the learnable functions is crucial for enhancing scalability. Third, rigorous benchmarking on large-scale, real-world datasets is essential to fully assess scalability and generalization. Finally, extending the KAN-AE framework to other data types and tasks—such as multi-modal representation learning, graph-structured data, or sequential data—would provide a comprehensive evaluation of its versatility and robustness, potentially unlocking novel applications across various scientific and industrial fields.

## 6 Conclusion

This paper introduced Kolmogorov-Arnold Network Autoencoders (KAN-AEs), a novel framework that integrates the adaptive, learnable function approximation of Kolmogorov-Arnold Networks into the autoencoder architecture. By replacing static activation functions with parameterized spline-based units, KAN-AEs capture complex data geometries more effectively than traditional autoencoders. We further proposed a Convolutional KAN-AE (CKAN-AE) variant, which incorporates translation equivariance and parameter sharing, making it particularly well-suited for image data.

Our comprehensive evaluation across benchmark datasets demonstrates that KAN-based autoencoders consistently achieve superior reconstruction fidelity and learn more discriminative latent representations, as measured by linear probing accuracy. The CKAN-AE variant proved especially powerful on natural images, balancing expressive capacity with the structural priors of convolutional design.

While the enhanced expressiveness of spline-based activations introduces a measurable computational cost, the significant gains in model performance establish a valuable trade-off for applications where precision and representation quality are critical. Future work will focus on optimizing the efficiency of these architectures and extending their principles to broader domains, including multi-modal and graph-structured data. Ultimately, this work underscores the potential of integrating foundational approximation theorems with deep learning to advance the state of unsupervised representation learning.

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
