# OpenReview forum: "Learning Expressive Representations with Kolmogorov-Arnold Network Autoencoders"
_TMLR — Rejected by TMLR_

### Review · Reviewer_QwG8 · 2025-12-02

**Summary Of Contributions:**

The paper proposes "Kolmogorov-Arnold Network Autoencoders" (KANA), a novel unsupervised learning framework that integrates Kolmogorov-Arnold Networks (KANs) into an autoencoder architecture. Unlike traditional autoencoders that use static activation functions (e.g., ReLU), KANA utilizes learnable, spline-based activation functions to capture complex non-linear relationships.The proposed method is evaluated against MLP-based baselines on the MNIST and Fashion-MNIST datasets, with results reported showing better reconstruction fidelity and latent space linear probing accuracy.

---

### Strengths
- Architectural Novelty: The paper presents an interesting extension of the recently proposed KAN architecture to the domain of dimensionality reduction and generative modeling.
- Comprehensive Ablation Studies: The authors conduct detailed ablations regarding encoder/decoder designs, hidden layer dimensionality, and loss functions (comparing MSE, MAE, Log-Cosh, etc.) to justify their architectural choices.
- Reconstruction Fidelity: On the limited datasets tested (MNIST/Fashion-MNIST), KANA demonstrates lower Mean Squared Error and higher linear probing accuracy compared to an equivalent MLP baseline.

### Weakness

- Mismatch Between Claims and Experimental Scale: The paper is titled "High-Dimensional Data Representation" and cites the "curse of dimensionality" as a motivation, yet experiments are restricted to low-resolution, low-dimensional datasets (MNIST and Fashion-MNIST). The absence of true high-dimensional benchmarks (e.g., CelebA, ImageNet) significantly undermines the central claim of the paper.
- Lack of Statistical Rigor: The experimental results report single scalar values for accuracy and MSE without error bars or standard deviations. Given the small scale of the datasets, multiple runs with different random seeds are necessary to verify that improvements are statistically significant and not due to initialization variance.
- Missing Computational Analysis: While the authors acknowledge "computational overhead" as a limitation, there is no quantitative comparison of training time, inference latency, or FLOPs against the baseline.
- Ineffectiveness of Proposed Convolutional Variant: The proposed "Convolutional KAN" (CKAN) actually underperformed the standard KAN in reported accuracy, raising questions about the scalability of the method to larger images where convolution is essential. A deeper analysis would be preferred to help the community.

**Additional Comments:**

I have described my concerns above.

**Audience:**

Yes

**Audience Explanation:**

I feel with more rigorous experimentation the paper can definitely benefit the audience.

**Broader Impact Concerns:**

I don't see any ethical concerns or implications here.

**Claims And Evidence:**

No

**Claims Explanation:**

The central claim of handling "High-Dimensional Data Representation" is unsupported, as the experiments are restricted to low-dimensional toy datasets (MNIST, Fashion-MNIST). Additionally, the claimed performance improvements are not convincing due to a lack of statistical rigor; the authors report single-run metrics without error bars or averages across multiple seeds, making it impossible to judge if the marginal gains are statistically significant. Finally, the "Convolutional" variant (CKAN), which is essential for scaling to high dimensions, actually underperformed the standard KAN on these simple tasks.

**Requested Changes:**

- I imagine training ImageNet can be compute intensive if there is lack of resources but if possible please include experiments on CelebA atleast. This is essential to demonstrate that the KANA framework scales beyond simple, low-dimensional toy datasets and to test the effectiveness of the proposed reconstruction capabilities on complex data.

- The current results (Tables 1-5) present single scalar values for MSE and Accuracy. Given the small scale of the current datasets, these margins could be due to initialization noise. Please repeat all key experiments (at least 3-5 runs with different random seeds) and report the Mean ± Standard Deviation. This is necessary to confirm that KANA consistently outperforms the baseline.

- Computational Efficiency Metrics: You mention "computational overhead" as a limitation but do not quantify it. Please add columns to your results tables comparing Training Time (wall-clock) and Inference Latency (or FLOPs) between KANA and the baseline MLP. The community needs to understand the cost-benefit ratio of using KANA.

- Currently, the baseline is an MLP. For image data, a Convolutional Autoencoder (CNN-AE) is the standard for high-fidelity reconstruction. Comparing KANA against a CNN-AE would provide a much stronger assessment of its utility in computer vision tasks.

- In Table 2, the CKAN encoder appears to underperform the standard KAN (87.2% vs 91.5% accuracy). Please provide a deeper analysis or discussion on why the convolutional variant which theoretically should handle spatial features better failed to improve performance. This is crucial for understanding the limits of the architecture.

---

### Review · Reviewer_APc6 · 2025-12-09

**Summary Of Contributions:**

This paper proposes KANA, which integrates KAN layers with learnable spline-based activation functions into standard autoencoder architectures. The authors evaluate their approach on MNIST and Fashion-MNIST datasets, demonstrating improvements in reconstruction MSE compared to MLP baselines.

Strengths:
1. Clear and accessible presentation of the KAN layer integration into autoencoders
2. Comprehensive ablation studies examining various architectural designs and hyperparameter configurations

Weaknesses:
1. The approach essentially replaces MLP layers with existing KAN layers in a standard autoencoder architecture, without substantial methodological innovation
2. The evaluation lacks experiments on genuinely high-dimensional, complex, or real-world datasets that would better validate the proposed approach
3. The paper lacks comparison with contemporary autoencoder variants such as VAE, $\beta$-VAE, VQ-VAE, and other state-of-the-art approaches

**Audience:**

No

**Audience Explanation:**

Applying existing KAN layers to standard autoencoders on MNIST is not sufficiently novel or insightful for a venue like TMLR. The community already knows learnable activations can improve neural networks, this paper doesn't advance that understanding meaningfully.

**Broader Impact Concerns:**

No significant ethical concerns.

**Claims And Evidence:**

No

**Claims Explanation:**

The experiments are limited to 28×28 grayscale images, which barely qualify as "high-dimensional data" by modern standards. The paper claims superiority for high-dimensional data processing but provides no evidence on actual high-dimensional datasets. The baseline comparisons are weak - only vanilla MLPs are tested, with no comparison to established autoencoder methods or recent representation learning approaches.

**Requested Changes:**

Evaluate on genuinely high-dimensional, modern datasets (ImageNet, CelebA, scientific datasets mentioned in intro)
Compare against relevant baselines: VAEs, other methods with learnable activations

---

### Review · Reviewer_d2J5 · 2025-12-14

**Summary Of Contributions:**

The paper introduces Kolmogorov-Arnold Network Autoencoders (KANA), an autoencoder framework. It integrates a spline-based representation to achieve a high degree of flexibility and expressiveness. It also proposes a Convolutional KAN (CKAN) layer:  kernel composed of learnable activation functions. KANA shows better reconstruction error and linear-probe accuracy than an MLP baseline.

**Audience:**

No

**Audience Explanation:**

"KAN: KOLMOGOROV–ARNOLD NETWORKS" (ICLR 2025) signifies that one of the most important aspect is interpretability. Hence, to be of interest to the TMLR community, I find it necessary to have experiments that demonstrates how KANA improves interpretability

**Claims And Evidence:**

No

**Claims Explanation:**

The author's claim "This deviation is motivated by the observation that the rigid 2n+1 structure may over-parameterize the activation
layer, leading to diminishing returns in performance". It is not clear how this claim is supported.

"Inspired by the architecture of CNNs, the Convolutional KAN (CKAN) layer introduces shift-invariant properties into neural networks, enabling the model to recognize patterns regardless of their spatial position.". Is this a contribution of this work? The idea behind CKANs have existed (https://arxiv.org/pdf/2406.13155). It's not clear.

**Requested Changes:**

1. Ablation on spline-based representation with a base activation function since it's a code contribution.
2. Experiments on interpretability wrt baselines.
3. Justify claims based above.

---

### Review · Reviewer_QdmJ · 2025-12-15

**Summary Of Contributions:**

The paper proposes using Kolmogorov–Arnold Networks (KAN, and a CKAN variant) as the building blocks of an autoencoder, in place of standard MLP layers. The authors motivate this with the Kolmogorov–Arnold representation theorem and test their KANA model on MNIST and Fashion-MNIST, reporting reconstruction and linear probing results, plus some small ablations.

**Audience:**

No

**Audience Explanation:**

KAN itself is a topic of current interest, but this paper mainly shows “replace MLP with KAN in a basic autoencoder on MNIST/Fashion-MNIST.” The novelty over existing KAN work is limited, the experimental setting is quite simple, and there are no strong insights (theoretical or empirical) that would substantially inform practice.

**Broader Impact Concerns:**

No special broader-impact concerns beyond those of standard representation learning methods; the current experiments are small-scale and not in sensitive domains.

**Claims And Evidence:**

No

**Claims Explanation:**

The main claims are quite broad (better for high-dimensional data, more expressive representations, wide applicability), but the evidence is limited to MNIST/Fashion-MNIST with fairly modest gains over a simple MLP autoencoder. There is no comparison to stronger and more standard baselines (e.g., convolutional AEs, VAEs, modern self-supervised methods), and the link to the Kolmogorov–Arnold theorem stays at the level of motivation rather than a concrete theoretical or empirical insight. Overall, the experiments are too narrow and the improvements too small to convincingly support the stronger claims.

**Requested Changes:**

1. Evaluate on more challenging and genuinely higher-dimensional datasets, and include stronger baselines (e.g., conv AEs, VAEs, standard SSL methods).
2. Clarify what is actually new beyond using existing KAN/CKAN layers in an AE, and either provide a more concrete theoretical angle or deeper empirical analysis.
3. Analyze representations more thoroughly (beyond small gains in linear probing), and report computational cost/overhead in more detail.
4. Tone down broad or “transformative” claims in the abstract and conclusion to match what is actually demonstrated.

---

### Decision · Action_Editor_Eo2U · 2026-02-09

**Recommendation:** Reject

**Additional Comments:**

The paper needs improvements to reach TMLR standards. First, include direct comparisons to at least one generative baseline (e.g., a standard VAE) to contextualize KAN-AE's performance within the broader autoencoder landscape. Deferring this entirely to future work is insufficient. Second, move beyond linear probing and reconstruction MSE by adding richer representation quality evaluations, such as downstream transfer tasks or latent space visualization analysis, to substantiate claims about learning "more expressive" representations.

**Audience:**

Yes

**Audience Explanation:**

Yes. KANs are a topic of active interest, and the paper offers useful empirical insights, particularly the ablations that could guide practitioners considering KAN layers. However, the core finding that swapping MLP layers for KAN layers modestly improves reconstruction is somewhat expected and doesn't yield a surprising insight. A narrow subset of readers interested in KAN architectures would find this informative, but appeal to the broader representation learning community remains limited.

**Claims And Evidence:**

No

**Claims Explanation:**

Not completely. The paper lacks comparison to stronger and more natural baselines like VAEs, β-VAEs, or VQ-VAEs. The authors acknowledge this but defer it to future work, which weakens confidence in the framework's practical value. The theoretical connection to the Kolmogorov-Arnold theorem remains motivational rather than substantive, so claims about the framework's expressiveness lack formal grounding. And while CIFAR-10 and STL-10 are a step up from MNIST, they're still relatively simple benchmarks that don't fully validate claims about learning expressive representations on complex data.

**Resubmission Of Major Revision:**

The authors may consider submitting a major revision at a later time.